

# Circulation of Baffin Bay and Hudson Bay waters on the Labrador Shelf and into the subpolar North Atlantic

Elodie Duyck[1], Nicholas P. Foukal[2], Eleanor Frajka-Williams[1],

[1] Institute of Oceanography, CEN, University of Hamburg, Hamburg, Germany
[2] Skidaway Institute of Oceanography, University of Georgia, Savannah, GA, USA

*Correspondence to*: Elodie Duyck (elodie.duyck@uni-hamburg.de)

**Abstract.** In the coming decades increasing amounts of freshwater are predicted to enter the subpolar North Atlantic from Greenland and the Arctic. If this additional freshwater reaches the regions where deep convection occurs, it could potentially dampen ventilation and the formation of deep waters. In this study we use a surface drifter dataset spanning the period 1990-
2023 to investigate the pathways followed by waters originating from Davis Strait and Hudson Strait on the Labrador Shelf and into the interior subpolar North Atlantic. Recent drifter deployments in the region allow for an improved understanding of the circulation on the Labrador Shelf, in particular its northern part, where prior data was sparse. We show that waters originating from Davis and Hudson Strait remain on the shelf as they flow downstream, until they reach the Newfoundland Shelf. This confirms that very little exchange take place between the Labrador Shelf and interior Labrador Sea. Decomposing
the Labrador Shelf into five regions, we further describe typical pathways for these waters and show that extensive exchanges take place between the coastal and shelfbreak branches of the Labrador Current. Our results suggest that if increasing amounts of freshwater reach the Labrador Shelf, it would not directly affect the Labrador Sea convection region; instead, it would lead to the formation a salinity anomaly off the Grand Banks, which could then circulate around the subpolar North Atlantic.

## 1 Introduction

In the interior seas of the subpolar North Atlantic, intense winter heat fluxes and relatively weak stratification allow for vertical mixing to depth reaching more than 1000 m (Lazier et al., 1980; Swift and Aagard, 1981; De Jong et al., 2012), leading to the formation of dense waters. This process, known as deep convection, allows for the ventilation of the intermediate to deep ocean, and contributes to the trapping of large amounts of anthropogenic carbon at depth (Steinfeldt et al., 2009; Rhein et al., 2017). Parts of the dense waters formed in these basins are then exported southwards as the lower limb of the Atlantic
Meridional Overturning Circulation (Buckley and Marshall, 2016; Buckley et al., 2023; Lozier, 2023).

In the coming decades, climate change is predicted to cause an increase in the amount of freshwater entering the subpolar North Atlantic, from Greenland (Bamber et al 2018; The IMBIE team 2020) and the Arctic (Haine et al. 2015; Shu et al. 2018). This increase would supplement interannual variations in freshwater input to the region, for instance related to changes in the Beaufort Gyre circulation (Timmermans and Toole 2023), recently suggested to be close to a change of state (Lin et al. 2023).



Increased freshwater fluxes to the subpolar North Atlantic could lead to a freshening of the upper layers in areas where deep convection usually happens, which could increase the stratification, and in turn weaken vertical mixing and deep water formation.

However, the freshwater that originates from Greenland and from the Arctic is first transported by narrow boundary currents over the continental shelves of Greenland and North America (Marsh et al., 2010). The freshwater must be mixed into the
interior before it can affect the stratification of deep convection regions. Freshwater pathways from the shelf regions to the deep convection regions are complex and there is no agreement on the volume of freshwater that would need to enter these regions to affect deep convection, nor how far in the future this could occur (Böning et al., 2016; Yang et al., 2016; Dukhovskoy et al., 2019).

In the Labrador Sea, most exchanges between the shelf and interior have been identified at the West Greenland Shelf, near
Cape Desolation. There, winds and eddies drive freshwater originating from Fram Strait and East Greenland glaciers into the central Labrador Sea (Lilly et al., 2003; Luo et al., 2016; Schulze Chretien and Frajka-Williams, 2019). In contrast, on the Labrador Shelf, little to no export of shelf waters into the interior is thought to take place (Myers et al., 2005; Penelly et al., 2019), with only very limited cross shelf exchanges described by observational studies (Howatt et al., 2018; Clément et al., 2023). While the waters exported near Cape Desolation originate from Fram Strait and the East Greenland glaciers, waters
found on the Labrador Shelf also originate from West Greenland glaciers (Gillard et al. 2016), the Arctic via Davis Strait (Lazier and Wright 1993), and Hudson Bay (Straneo and Saulcier 2008). To understand the possible impact of increased freshwater fluxes from Greenland and the Arctic on the subpolar North Atlantic it is therefore essential to understand how the waters that enter the Labrador Shelf circulate over the shelf, and where and how they leave it to enter the open ocean.

Our understanding of the circulation on the Labrador Shelf dates back from early oceanographic expeditions of the late 19th
and early to mid-20th century (Smith 1937) and was informed by repeated summer sections along the shelf, as well as mooring arrays from the late 20th century onwards (Lazier and Wright, 1993; Cyr and Galbraith, 2021). At the northern end of the Labrador Shelf, the Baffin Island Current transporting Baffin Bay and West Greenland Current waters (Cuny et al., 2005; Curry et al., 2014), the Hudson Strait outflow (Straneo and Saulcier, 2008; Ridenour et al., 2021), and the remaining waters from the West Greenland Current, come together to form the Labrador Current (Fig. 1a). The Labrador Current is composed
of three branches (Lazier and Wright 1993; Fratantoni and Pickart, 2007): (1) an outer, barotropic branch, found around the 1500m isobath; (2) a main, surface intensified branch, centred at the 500m isobath; and (3) an inner branch, that closely follows the coast, also referred to as the Labrador Coastal Current. In the following we refer to the three branches as "slope", "shelfbreak" and "coastal" Labrador Current. The shelfbreak Labrador Current is typically indistinguishable from the slope Labrador Current in velocity sections but the water masses it transports are colder and fresher (Lazier and Wright, 1993;
Fratantoni and Pickart, 2007). While the coastal Labrador Current can be distinguished as a separate branch, its origins are



disputed, with literature both suggesting it stems from the Hudson Strait outflow (Florindo Lopez et al., 2020), or from the steering of waters into the deep troughs and canyons that cut across the shelf (Peterson, 1987; Colbourne et al., 1997; Fig. 1b).

As it reaches Newfoundland, the strong coastal current bifurcates towards the shelf edge to join the shelfbreak current. A limited part of the coastal current remains close to the coast and flows through the strait of Belle Isle, north of Newfoundland, or towards the Avalon Channel, east of Newfoundland (Petri and Anderson 1983). At the shelf edge, the Labrador Current splits as it reaches Flemish Cap: one branch enters Flemish Pass, while another rounds Flemish Cap. Between Flemish Cap and the tail of the Grand Banks, the Labrador Current is retroflected towards the subpolar North Atlantic and loses approximately 90% of its volume transport (Lazier and Wright, 1993; Loder et al., 1998; Fratantoni and Pickart, 2007). The strength of the retroflection, and therefore the amount of waters diverted from the Labrador Shelf into the subpolar North Atlantic varies on seasonal and interannual timescales, modulated by the large-scale wind patterns and the strength of the Labrador Current (Petrie and Drinkwater, 1993; Jutras et al., 2023; Goncalves Neto et al., 2023).

The goal of this study is to characterise the pathways followed by freshwater originating from Baffin and Hudson Bay on the Labrador Shelf and into the subpolar North Atlantic, and to better understand the possible downstream impact of increased freshwater fluxes from these regions. We investigate these pathways using surface drifters. In Sect. 3.1, we describe the surface circulation as inferred from the drifter dataset and compare it with existing studies. In Sect 3.2, we investigate exchanges between the inner Labrador Shelf, where the fresh waters are found, and the central Labrador Sea, where deep convection takes place. In Sect 3.3, we define the typical pathways of drifters originating in Baffin and Hudson Bay over the Labrador Shelf. In Sect 3.4, we investigate where these drifters enter the interior ocean and how they spread across the subpolar North Atlantic. In Sect 4, we discuss what these freshwater pathways imply for the future influence of waters from Baffin Bay and Hudson Bay on deep convection in the subpolar North Atlantic.



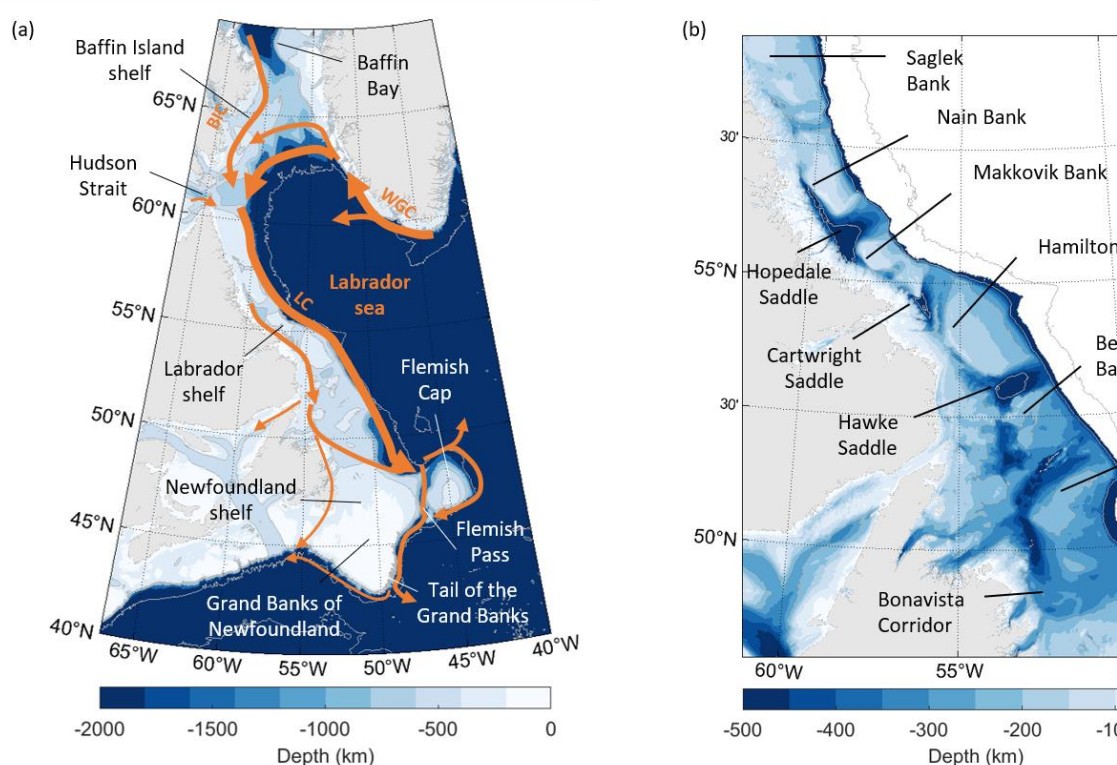

**Figure 1: (a) Surface circulation around the Labrador Sea and on the Labrador Shelf. WGC: West Greenland Current, BIC: Baffin Island Current, LC: Labrador Current. (b) Topographic features along the Labrador Shelf.**

## 2 Data and methods

### 2.1 Surface drifters

We use surface drifter trajectories from the Global Drifter Program (GDP) 6-hour interpolated data set (Centurioni et al., 2019; Lumpkin & Centurioni, 2019), and from the East Greenland Current Drifter Investigation of Freshwater Transport dataset (EGC-DrIFT, Duyck and De Jong, 2023). From the GDP, we use all available data from January 1990 to May 2023. From EGC-DrIFT, we use Surface Velocity Program drifters that are similarly interpolated every 6 hours, between August 2019 and November 2022 (Duyck and De Jong, 2023). The Surface Velocity Program drifters from both datasets are fitted with a holey sock drogue, centred at 15 m depth, which ensures limited wind drift, and allows them to follow the circulation at that depth. We filter the drifter trajectories with a 25 h Butterworth filter to remove high frequency signal due to tides and inertial oscillations and recompute drifter velocities from the filtered trajectories. To ensure the quality of the dataset, we



exclude drifters that have lost their drogues, and cut the trajectories of drifters that grounded ashore or did not transmit for more than 30 days in a row. This is notably the case for one of the drifters shown in Clément et al. (2023), that appears to have been trapped under sea ice and was exported away from the shelf shortly after resurfacing. The result of this filtering is shown Fig. S1.

Since 2019, several drifter deployments took place at the Greenland shelf, motivated by the growing interest in the interactions
between fresh continental shelves and the deep convection regions. While a few studies already investigated surface circulation at the Labrador Shelf using drifters (Reverdin et al., 2003, Cuny et al., 2002, Fratantoni et al., 2001), they were limited regionally due to the paucity of drifter trajectories on the whole shelf. Even with the new data collected since 2019, the drifter coverage of the region is still heterogeneous, with an order of magnitude more drifter data available over the central Labrador Sea compared to the Labrador Shelf (Fig. 2a). However, the large increase in available data over the whole region also included
the northern Labrador Shelf, where little to no drifter data were previously available (Fig. 2b).

While these new data allow for new analysis of the circulation on the Labrador Shelf, the resulting time concentration of the dataset can lead to bias: Out of 35 drifters that passed through Davis Strait, 25 were deployed at the same time, in September 2021, as part of the TERIFIC (Targeted Experiment to Reconcile Increased Freshwater with Increased Convection) experiment. Moreover, sea ice cover prevents drifters from flowing over most of the Labrador Shelf in winter and spring, and
may also affect their behaviour in the late autumn: Part of the TERIFIC drifters mentioned above were caught in sea ice as they flowed over the Labrador Shelf, resulting in a lower amount of available data in late winter and early spring over the Labrador Shelf (Fig. 2c).





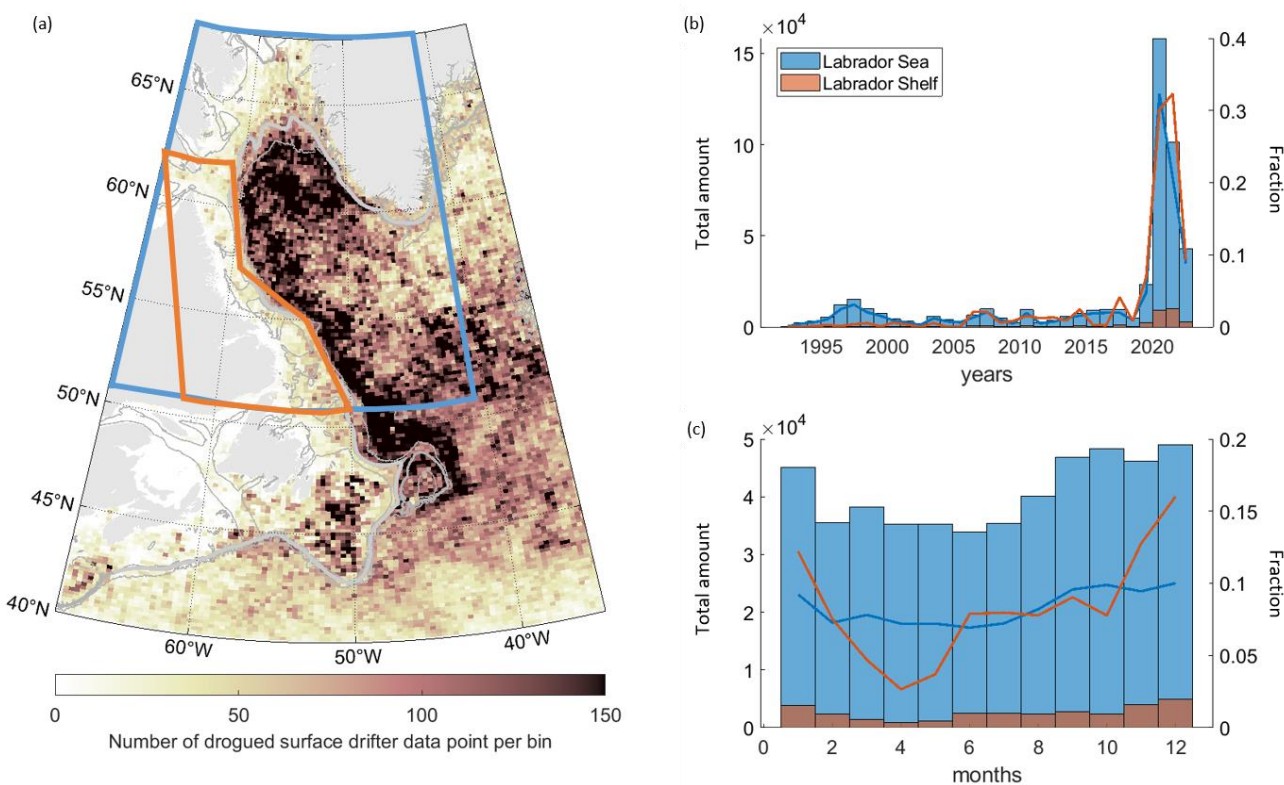

**Figure 2: (a) Number of drifter data points per bin. Bins are 1/3° latitude per 1/6° degree longitude. Bathymetry in grey: 2000 m,**
**1000 m, 250 m isobaths. (b) Total amount (bar plot) and fraction (line plot) of drifter data points per year in the Labrador Sea area**
**(blue) and Labrador Shelf area (Red) as defined in (a). (c) Same for each month.**

## 2.2 Definition of the shelf boundary and drifter crossing computation

To identify cross shelf exchanges in the Labrador Sea, we define a shelf boundary using the 1000 m isobath. The choice of
this isobath is motivated by the need for a boundary that works well for both the West Greenland and Labrador Shelf, the
former having a steeper slope than the latter. We smooth the shelf boundary with a 150 km window and interpolate it to a
resolution of 5 km. The resulting shelf boundary is shown Fig. 3a, together with the along shelf distance from Cape Farewell.
To detect where and when drifters cross into or out of the shelf region, we divide the shelf boundary in 200 km segments and
compute the number of crossings across each segment. Some drifters cross the shelf boundary multiple times. To identify
locations of net offshore crossings we use the following criteria: if the drifters were on the shelf prior to crossing the segment
of the shelf boundary, crossed it in the offshore direction more often than in the inshore direction, and did not cross back



inshore at the five following segments (which corresponds to 750km), they are considered as exported at that segment. These criteria also allow to avoid the complex shape of the shelf boundary from affecting the results.

## 2.3 Drifter pathways over the Labrador Shelf

Of the 35 drifters that entered the Baffin Island shelf via Davis Strait, only five reached the Newfoundland Shelf while their drogues were still attached. The rest lost their drogues as they flowed over the Labrador Shelf, were stranded at the coast, or were caught into sea ice. Instead of only considering drifters that flowed all the way from Baffin / Hudson Bay to the Newfoundland region, we divide the region into five section, and investigate the behaviour of drifters between each of these sections. This section-to-section analysis increases the number of drifters we can include in our study, which limits time bias

and enables the drawing of more robust conclusions. We define five sections, as cross-shelf sections (e.g., from the Labrador coast towards the interior Labrador Sea) or as gateways across topographic constrictions (e.g., Davis Strait).

The five sections are further divided into a total of 10 boxes, defined manually to differentiate between drifters flowing in the coastal and shelfbreak branches of the Labrador Current (Fig. 3b). The sections and boxes are referred to as: (1) The Davis Strait section, (2) the northern Labrador Shelf section, at the mouth of Hudson Strait, with an inner and outer box to separate

the Baffin Island Current and the remnant of the West Greenland Current, (3) the mid- Labrador Shelf section at 56°N, with an inner and outer box to detect drifters steered towards the coast along the canyons that cut across the Labrador Shelf, (4) The southern Labrador Shelf section, just south of Hamilton bank, with an inner and outer box to differentiate the shelfbreak and coastal current, (5) the Newfoundland Shelf section, with three boxes to identify drifters rounding Flemish Cap, channelled through Flemish pass, or flowing over the shallow Newfoundland Shelf and the strait of Belle Isle.

To resolve the pathways of drifters flowing over the Labrador Shelf, we identify drifters that flow from one section to the next and separate them depending on the box they originated from and arrived to at each section. If the drifters passed through several of the boxes of the same section, we consider the last box it passed through before proceeding to the next section, and the first box it passed through after arriving to the section. From this, we can build an overview of the number of drifters flowing from one box to the next and preferred pathways, as well as the median travel time.

This method allows us to define typical pathways with a much larger drifter dataset than if only considering those that flowed all the way from Davis Strait to the Newfoundland Shelf. Instead of only these five drifters, we use data from 171 different drifters, that flowed across two or more of the defined sections. The additional drifters were either deployed downstream of Davis Strait, recirculated onto the Baffin Island or Labrador Shelf after flowing with the West Greenland Current, and / or stopped working before they reached Newfoundland.





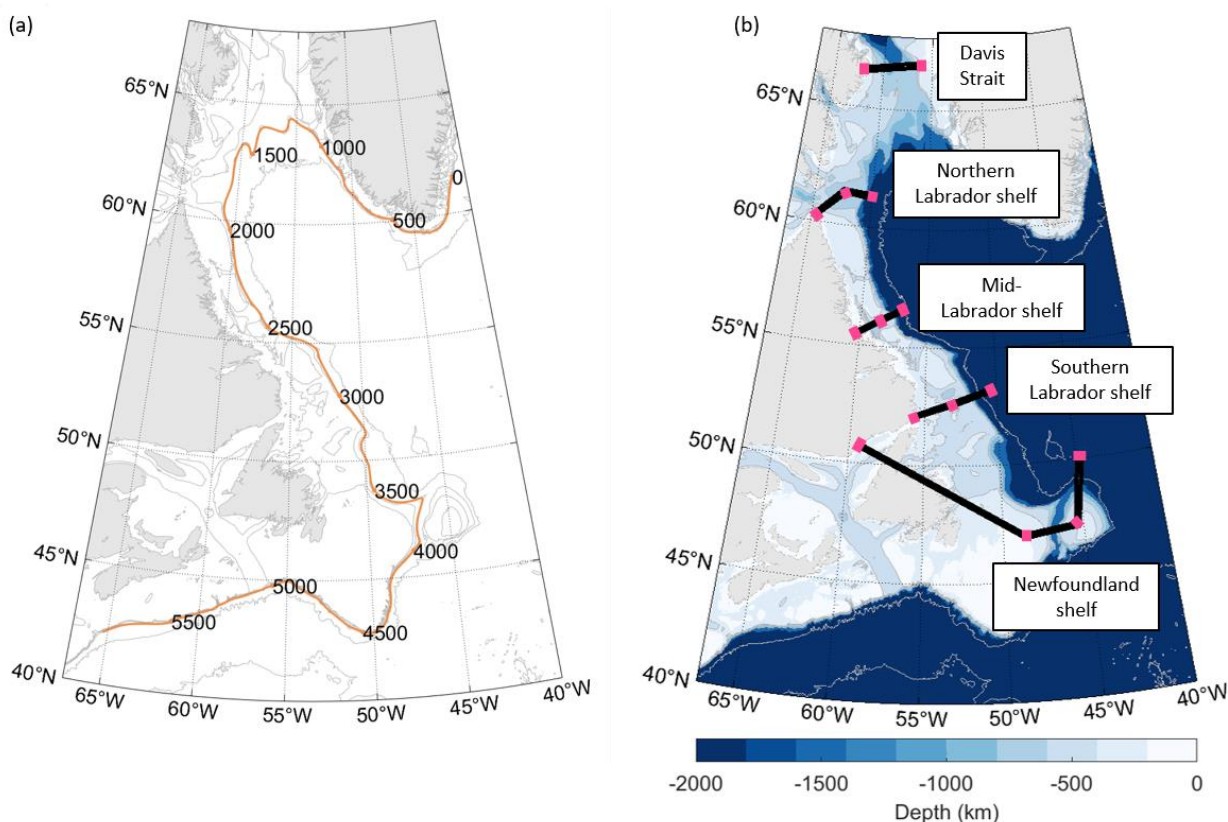

**Figure 3: (a) Definition of the shelfbreak as the smoothed 1000m isobath, and along-shelf distance from Cape Farewell in km. Bathymetry in grey: 5000 m, 2500 m, 1000 m, 500 m, 250m isobaths. (b) The five sections and 10 boxes defined over the Labrador Shelf to identify typical pathways for waters flowing from Davis Strait and Hudson Strait to the Newfoundland region.**

## 3. Results

### 3.1 Surface circulation over the Labrador and Newfoundland shelves

In this section we build a map of the surface circulation from the drifter datasets to identify the main characteristics of surface currents in the Labrador Sea, and compare the drifter-inferred circulation with existing knowledge, which was developed mostly from synoptic sections, moorings, and models. Using the full drifter dataset, from 1993 to 2023, we construct a pseudo-Eulerian map of the surface circulation in the Western subpolar North Atlantic (Fig. 4). The map is built as a 1/3° latitude per 1/6° degree longitude bins, with the velocity at each bin computed as the average velocity of all drifter data available in the region corresponding to that bin. Bins are only shown if they represent more than five data points, from at least two separate drifters.





The main export of shelf waters into the Labrador Sea takes place at the West Greenland Shelf, between Cape Desolation (60°N) and Fylla Bank (63°N). This is visible in Fig. 4 as a broadening and dissipating of part of the West Greenland Current into the Labrador Sea. Past Fylla Bank, three distinct branches of the West Greenland Current are distinguishable: (1) part of the West Greenland Current is steered westwards, along the 1500 m to 2500 m isobaths. (2) Another part continues northwards along the 500 m isobath and recirculates into the Baffin Island Current as it reaches Davis Strait, (3) A small portion of the flow continues northwards along the West Greenland Shelf, into Baffin Bay.

The Baffin Island Current is composed both of waters originating from Baffin Bay, and waters recirculating from the West Greenland Current. In addition to the main branch of the Baffin Island Current, situated at the 500m isobath, the drifter dataset shows a secondary flow close to the Baffin Island coast, which could be a coastal branch of the Baffin Island Current. However, the drifters which contributed to the pseudo-Eulerian map in this coastal flow were all deployed in September 2021 so from these data we cannot conclude whether it is a permanent feature (Fig. S2a shows more clearly the drifters that circulate in the main branch of the Baffin Island Current, and closer to the coast).

As it approaches Hudson Strait, the Baffin Island Current is steered towards its entrance. Part of the drifters found in the Baffin Island Current entered the Hudson Strait for a short time, before recirculating back onto the shelf. Others, that flowed inshore of the main branch of the Baffin Island Current, entered Hudson Strait by its northernmost end and continued upstream into the strait. This is consistent with Ridenour et al. (2021) and with the drifters analysed in Le Blond et al. (1981, not included in this dataset), that both show that the Hudson Strait inflow is of Baffin Bay origin and occurs north and south of Resolution island.

Past Hudson Strait, the Hudson Strait outflow, the Baffin Island Current and the remainder from the West Greenland Current combine at the Labrador Shelfbreak to form the Labrador Current. Only one core is visible at the shelfbreak, in agreement with the description of the Labrador Current from Lazier and Wright (1993). The available drifter data does not allow detection of a coastal branch of the Labrador Current north of 58°N. There is no clear connection between the Hudson Strait outflow and the Labrador Coastal Current directly past Hudson Strait: Of the 11 drifters that passed through Hudson Strait, only two drifters remained close to the coast as they entered the Labrador Shelf. One of them beached shortly afterwards, while the other lost its drogue after being trapped in sea ice. The other nine drifters all joined the main branch of the Labrador current at the shelfbreak (Fig. S2b).

Over the Labrador Shelf, deep canyons drive exchanges between the shelf edge and the inner shelf. The Labrador Shelf circulation map (Fig. 4b) shows three such features, at 58 N, 56 N and 55°N, between Salgek and Nain Banks, at the Hopedale Saddle, and at the Cartwright Saddle (See Fig. 1b for names of topographic features). In all cases, some of the drifters steered towards the coast were entrained in the coastal branch of the Labrador Current, while the remainder were steered back towards the shelfbreak branch. This is very similar to the circulation that Peterson et al. (1987) inferred from using ice floes. The coastal



branch of the Labrador Current is not visible along the whole Labrador Shelf in the pseudo-Eulerian map, which is likely a

200    consequence of the sparse drifter coverage over the inner shelf.





**Figure 4: (a) Pseudo-Eulerian map of the surface circulation in the western subpolar North Atlantic. Each bin shows the average drifter speed in a 1/3° latitude per 1/6° degree longitude region. Bins are shown only if they represent more than 5 data points, and if these data point are from at least two different drifters. The bathymetry in white shows the 2500 m, 1000 m and 500 m isobaths. (b) Same as (a), for the Northern Labrador Shelf. (c) Same as (a), for the Newfoundland shelf.**

Past Hamilton Bank, from 53°N, several pathways are visible from the shelfbreak over the shelf and into the coastal current, through the Hawke Saddle and Funk Island Deep. This enhanced exchange is consistent with previous work that showed drifters steered towards the coast in that region (Colbourne et al. 1997, not included in this dataset) and identified an increase in West Greenland Current origin waters over the shelf past Hamilton Bank (Benetti et al. 2017).

The circulation map shows three possible pathways for waters found in the coastal current as it reaches the Newfoundland shelf, though only the main one is clearly visible: (1) A few of the drifters are steered into the Strait of Belle Isle or (2) continue along the Newfoundland coast into the Avalon Channel (Petrie and Anderson 1983), while (3) the majority of the drifters are channelled through the Bonavista Corridor towards the shelfbreak Labrador Current.

As it reaches Flemish Cap the Labrador Current splits. Part of the current is driven away from the shelfbreak and flows around Flemish Cap. The rest remains at the shelfbreak and flows through Flemish Pass. Between Flemish Pass and the Grand Banks, drifters are diverted away from the shelfbreak and entrained into the North Atlantic Current (Fig. 4c). This is clearest at the tail of the Grand Banks. Past the tail of the Grand Banks, no fast current is distinguishable at the shelfbreak anymore, consistent with the retroflection of the Labrador current identified in that region (Fratantoni and Pickart 2007, Jutras et al 2023).

**3.2 Cross-shelf exchanges into the central Labrador Sea**

In this section, we investigate cross shelf exchanges between the continental shelves and central Labrador Sea, and the origin of drifters exported in the Labrador Sea. We determine the regions where most exchanges occur by dividing the shelf boundary in 200 km along-shelf segments and identifying drifters that were exported from the shelf into the interior across these segments, as defined in Sec. 2.2 (Fig. 5).

There are two main regions where a substantial fraction of the drifters found on the continental shelves of the Labrador Sea are exported (Fig. 5a-b): (1) the eddy shedding region of the West Greenland Shelf, and (2) Flemish Pass and the Grand Banks. Past the Grand Banks, the number of drifters found over the shelf strongly diminishes due to the retroflection of the Labrador Current. A minority of drifters also leave the shelf in the region just before Flemish Pass. In the following, we focus on exchanges taking place between the Labrador and Newfoundland shelves, and the open ocean.

Example trajectories for drifters exported across groups of segments of the shelf boundary show that not only different areas of the shelf have less or more exchanges, but also it is partly drifters of different origins that are exported in different areas: the drifters that crossed the shelf boundary from the Labrador Shelf into the central Labrador Sea before reaching the Newfoundland shelf nearly only originated from the West Greenland Current (Fig. 5c-d). By contrast, the drifters exported



past Flemish pass originated from both the West Greenland Current and from Baffin and Hudson Bay, with part of these drifters circulating over the inner shelf before they reached Flemish Pass (Fig. 5e).

These results suggest a distinction in the origin and fate of waters transported by different branches of the Labrador Current: Waters transported in the slope Labrador Current, that originate from the West Greenland Current, remain at the offshore edge of the shelf and can be exported into the central Labrador Sea. These are however not particularly fresh waters. Waters transported by the shelfbreak and coastal Labrador Current, which notably includes very fresh waters of Hudson and Davis Strait origin, only leave the shelf further downstream, around Flemish Cap and the Grand Banks.










**Figure 5 (a) Number of drifters exiting the shelf at a given segment of the shelf boundary, divided by total number of drifters found over the shelf at that segment (within 100km of the shelf edge). Drifters are not considered if they immediately re-enter the shelf at the next segment. (b) Total number of drifters (pale orange) and number of drifters leaving the shelf (red) at a given segment of the shelf boundary. (c)(d)(e) Three examples of drifter trajectories before (blue) and after (red) they leave the shelf at groups of segments (shown in black). For all panels, bathymetry in grey: 5000 m, 2500 m, 1000 m, 500 m, 250 m isobaths.**


To further this analysis, we look specifically at how drifters originating from Davis Strait and Hudson Strait flow over the Labrador Shelf and enter the subpolar North Atlantic (Fig. 6). Drifters from these two origins, shown respectively in red and in blue, display a similar behaviour. The large majority of the drifters remained over the shelf until they reached the

Newfoundland shelf, and was only exported into the subpolar North Atlantic at, and past, Flemish Pass. Only one of the drifters crossed the shelf boundary into the Labrador Sea (Blue trajectory, between 56°N and 52°N), and it remained in the Labrador Current downstream. We also remark that some of the Hudson Strait drifters spread over the inner Labrador Shelf earlier than Davis Strait drifters, though it does not seem to lead to differences in their downstream trajectories.

Both the analysis of cross-shelf exchanges along the Labrador Shelf, and of the trajectories of drifters originating from Davis

/ Hudson Strait suggest little to no direct connection between Baffin / Hudson Bay and the interior Labrador Sea, with waters from these origins only exported downstream of Flemish Pass and Flemish Cap. On the other hand, the spreading of drifters over the shelf suggests extensive exchanges between the shelfbreak branch of the Labrador Current its coastal branch.

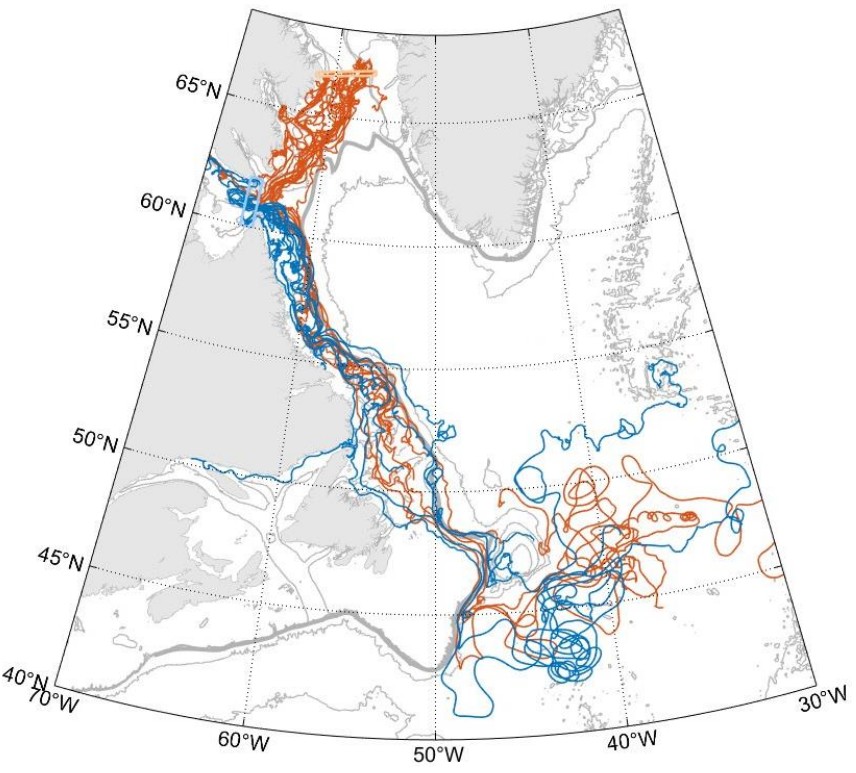



**Figure 6: Trajectories of drifters originating from or passing through Davis (red) and Hudson (blue) straits. Drifters that originate**
**from Davis Strait but also enter and exit Hudson Strait are marked as Hudson Strait drifters. The boxes used to identify drifters**
**crossing the straits are shown in light red and light blue. Bathymetry in grey: 5000 m, 2500 m, 1000 m, 500 m, 250m isobaths.**

### 3.3 Pathways of Baffin Bay and Hudson Bay waters over the Labrador Shelf

Only a few drifters were found both at Davis / Hudson Strait and at the Newfoundland Shelf. In the following we investigate
how drifters circulate over the shelf by considering their trajectories between the five sections defined in Sect 2.3.

Between the Davis Strait and northern Labrador Shelf section, drifters mostly remain over the Baffin Island shelf: nearly all
(11 out of 12, Fig, 7) of the drifters originating from the Davis Strait section cross the inshore box of the northern Labrador
Shelf section (Fig. 8a). These drifters are joined at that inshore box by some of the drifters that recirculated from the West
Greenland Current into the Baffin Island Current, as well as drifters from the Hudson Strait outflow.

Between the inshore box of the northern Labrador Shelf section and the southern Labrador Shelf section, the drifters largely
remain on the shelf but display strong exchanges between the shelfbreak and coastal branches of the Labrador Current (Fig.
8b). Drifters originating from the inshore box of the northern Labrador Shelf section first join the Labrador Current at the
shelfbreak. Most are then steered towards the coast via the canyons that cut across the Labrador Shelf: At the mid- Labrador
Shelf section, 86% of these drifters are found in the inshore box, which corresponds to the coastal Labrador Current and to the
circulation along the topography of the canyons. While 61% of the drifters found in the inshore box at the mid-Labrador Shelf
section remain in the coastal current downstream until the southern Labrador Shelf section, the other 39% are driven back
towards the shelf edge and found in the shelfbreak Labrador Current by the southern Labrador Shelf section (Fig. 7).

By contrast, between the offshore box of the northern Labrador Shelf section and the mid-Labrador Shelf section, drifters
remain at and off the shelfbreak. Limited exchanges with the inner shelf only take place between the mid and southern Labrador
Shelf sections, past Hamilton Bank (Fig. 8c). The large majority (89%) of the drifters found in the offshore box at the northern
Labrador Shelf section are also found in the offshore box at the mid Labrador Shelf section. Between the mid and southern
Labrador Shelf sections, 17% of the drifters originally found at the shelfbreak enter the inner shelf and are found in the onshore
box of the southern Labrador Shelf section (Fig 7). This behaviour is shown by the drifter trajectories drawn in yellow in Fig.
8c, that remain at and off the shelfbreak upstream of Hamilton Bank and only spread over the shelf from and downstream of
Hamilton Bank. The one drifter from Davis Strait that crosses the offshore box at the northern Labrador Shelf section follows
this latter pathway.

In summary, until 54°N, the drifters found at the offshore box at the northern Labrador Shelf section mostly remain at the
shelfbreak and off the shelf, while drifters found at the inner box spread over the inshore shelf. Past 54°N, drifters that were
found at the inshore box continue showing exchanges between the coastal and shelfbreak branch, but part of the drifters that
originated from the offshore box also take part to these exchanges.




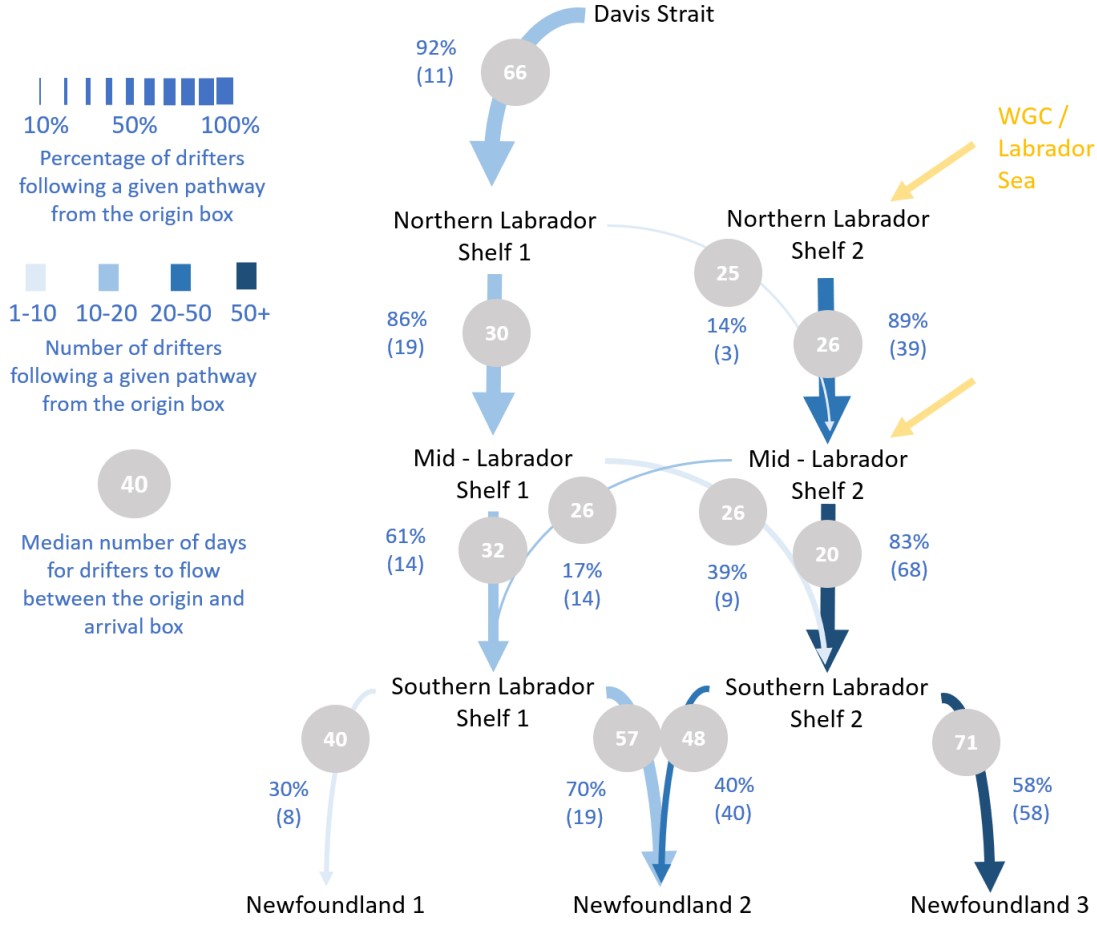

**Figure 7: Pathways followed by drifters between Davis Strait and Newfoundland. All computations correspond to drifters flowing from a given origin box of a first section, to a second section, and therefore only consider drifters present at both sections. The width of the arrows indicates the percentage of drifters following a given path from the origin box, while the intensity of the colour indicates the number of drifters following that path. The grey circles show the median number of days drifters took to go from a box to the next. Only paths that are followed by more than 10% of drifters from a given box are shown.**

Between the inshore box of the southern Labrador Shelf section, which corresponds to the coastal Labrador Current, and the Newfoundland section, three distinct behaviours can be observed (Fig. 8d). The drifters that remain at the inner shelf and cross the Newfoundland box at the Newfoundland section (30% of the drifters found in the coastal current at the southern Labrador Shelf section) either enter the strait of Belle Isle or flow along the Newfoundland coast towards the Avalon Channel. The remaining 70% flow towards the shelfbreak and are found at the Flemish Pass box at the Newfoundland section. None of the drifters originating from the inshore box of the southern Labrador Shelf section were found at the Flemish Cap box at the



Newfoundland section, confirming that drifters found in the coastal current over the southern Labrador Shelf can only exit the shelf downstream of Flemish pass.

Between the offshore box of the southern Labrador Shelf, which corresponds to the shelfbreak and slope Labrador Current, and the Newfoundland section, drifters either flow through Flemish pass (40%) or circulate around Flemish Cap (58%). Only two drifters crossed the Newfoundland box, as they flowed over the Newfoundland shelf slightly west of Flemish Pass. Fig. 8e suggests that the drifters that circulated around Flemish Cap never circulated over the Labrador Shelf (trajectories in red). By contrast, all the drifters that at some point circulated over the shelf were driven into Flemish pass (in blue), with a small

minority going over Newfoundland shelf (in green).

This is confirmed when also investigating trajectories directly between the northern Labrador Shelf and Newfoundland sections (Fig. S3 and S4): Only 12 of the drifters found at the inshore box of the northern Labrador Shelf section were still active when they reached the Newfoundland section. Of these, 10 crossed the Flemish Pass box, with the two others crossing the Newfoundland box after flowing through the Strait of Belle Isle and along the Newfoundland coast. On the other hand, of the

33 drifters that originated from the offshore box of northern Labrador Shelf section and are still active at the Newfoundland section, 37% circulated around Flemish Cap, while 57% flowed through Flemish Pass.

These results suggest that waters from Hudson and Davis Strait, found at the inner shelf upstream of the Labrador Shelf, remain over the Labrador Shelf as they flow downstream, with large exchanges between the inner part of the shelf and its edge. These water masses only enter the open ocean downstream of Flemish Pass. When adding the times taken by drifters to flow between

the different sections, we find that they take a median time of 185 days from Davis Strait to Flemish Pass if they flow in the coastal branch of the Labrador Current over the shelf, and 160 days if they flow in the shelfbreak current (Fig. 7).










**Figure 8: Trajectories of drifters between (a) The Davis Strait and northern Labrador Shelf sections. (b)(c) The northern and southern Labrador Shelf sections. (d)(e). The southern Labrador Shelf and Newfoundland sections. The black box shows the origin box of the drifters displayed, while the coloured boxes show the different boxes of the end section. Drifters are coloured according to which box of the end section they cross first. The bathymetry is shown in grey with the 5000 m, 2500 m, 1000 m, 500 m, 250m isobaths, and the shelf boundary is shown as a thick grey line.**

### 3.4 Export of surface drifters at Flemish Cap and the Grand Banks

In this section, we investigate where drifters are exported from the shelf after passing Flemish Pass and Flemish Cap, and how they then circulate around the subpolar North Atlantic.

The drifters that flowed through Flemish Pass exit the Labrador Current between Flemish Pass and past the Tail of the Grand Banks. To investigate where more exactly they leave the shelf and are entrained into the interior subpolar North Atlantic, we use the along shelf distance from Cape Farewell (as defined Sect. 2.2) to identify where drifters stop following the shelf boundary and retroflect. We use three along-shelf distances (Fig. 9): 4000 km, at the southern end of Flemish Pass, 4300 km, upstream of the Tail of the Grand Banks, and 4600 km, downstream of the Tail of the Grand Banks.

The drifters that stopped following the shelf before and at the southern end of Flemish Pass (23 of the 104 drifters that crossed Flemish Pass) were entrained in the anticyclonic circulation around Flemish Cap or bifurcated north eastwards at the sharp bathymetric bend south of Flemish Pass (Fig. 9a). These drifters entered the interior ocean south of Flemish Cap.

Drifters that reached the Grand Banks, and stopped following the shelf before the tail of the Grand banks, bifurcated north-eastwards over that whole area (Fig. 9b), in a similar way as the drifters exported at the tail of the Grand Banks (Fig. 9c). This shows that the retroflection of the Labrador Current takes place from the southern end of Flemish Pass to the Tail of the Grand Banks, even as it seems to be intensified in that latter region. This is consistent with previous studies of the retroflection, such as Fratantoni and Pickart (2010) and Goncalves Neto et al. (2023), who also show that the retroflection area can additionally vary over time.

In our dataset, all the drifters that were still found in the Labrador Current past the Tail of the Grand Banks, had left the shelf by the Laurentian Channel. After crossing the shelf boundary, these drifters flowed eastwards and later north-eastwards, following a similar trajectory than the drifters retroflected at the Grand Banks (Fig. 9d). This is in contrast with earlier work, as for instance Jutras et al. (2023) showed that 25% of the particles transported in the Labrador Current continued downstream over the Scotian Shelf and Gulf of Maine between 1993 and 2015. This difference could be due to a temporal bias in the drifter dataset toward more recent years, as well as a heterogeneous seasonal coverage, with four times more drifters found over the Newfoundland shelf in August than February (Goncalves Neto et al., 2023).



The drifters that left the shelf before the Flemish Pass box, and circulated around Flemish Cap, also exhibit a range of behaviours, illustrated Fig. S5. There are two main areas where drifters left the circulation around Flemish Cap: In the northern part of Flemish Cap only drifters that were found past the 2500 m isobath at the Flemish Cap box were exported. Most drifters were exported in the south-east and southern part of Flemish Cap, where the bathymetry is the steepest. This is notably the case of the drifters that rounded Flemish Cap above its shallowest parts. The drifters leave the circulation around Flemish Cap in the same regions (northeast and south of Flemish Cap) than Solodoch et al. (2020) described for the Deep Western Boundary Current.





**Figure 9: Behaviour of drifters after they flow through the Flemish Pass box. The panels show drifters detrained from the Labrador Current (a) before reaching the southern end of Flemish Pass (4000 km along-shelf from Cape Farewell); (b) between the southern end of Flemish Pass and the Tail of the Grand Banks (4000-4300 km); (c) at the Tail of the Grand Banks (4300-4600km); (d) past the Tail of the Grand Banks (beyond 4600km along shelf from Cape Farewell). In all cases, the darker blue trajectories represent examples of trajectories for the group of drifters. The bathymetry is shown in grey with the 5000 m, 2500 m, 1000 m, 500 m, 250 m isobaths.**

The upstream origin of the drifters influences where they are exported in the Newfoundland-Flemish Cap region. While part

of the drifters found in the slope and shelfbreak Labrador Current round Flemish Cap seaward and are entrained into the North



Atlantic Current northeast of Flemish Cap, drifters originating from the coastal current do not leave the shelfbreak before they
reach the southern end of Flemish Pass and the Grand Banks (Fig. 10). This is also the case more generally for drifters coming
from Davis Strait and Hudson Strait, as shown in the previous section. However, even as there is initially a strong difference
in where the drifters are detrained from the Labrador Current and entrained into the North Atlantic Current, this difference
disappears as they spread into the subpolar North Atlantic (Fig. 10). All the drifters exported between the northern part of
Flemish Cap and past the Tail of the Grand Banks spread north eastwards into the subpolar North Atlantic, along the path of
the North Atlantic Current.

We find that it takes a median time of 206 days, nearly seven months for drifters to reach 30°W from the Flemish Pass section
(212 from Flemish Cap). In the previous section, we showed that it takes a median time of 185 days for drifters to flow from
Davis Strait to Flemish Pass. According to this dataset, it would therefore take more than a year for an anomaly to spread from
Davis Strait to the mid-Atlantic ridge.

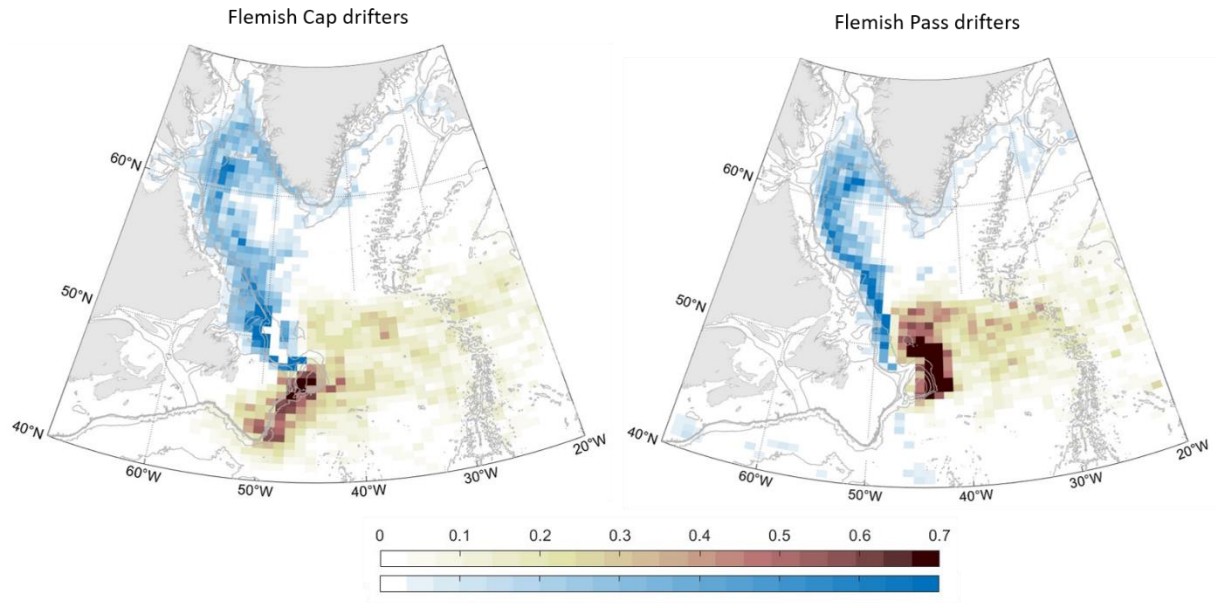


**Figure 10: Distribution of drifter data from the drifters that passed through the Flemish Pass (left) and Flemish Cap (right) boxes.
For both Flemish Cap and Flemish Pass drifters, the coloured bins show the percentage of the total number of data points from
these drifters found in a given bin, with in blue the data collected before the drifters crossed the box, and in red after the drifter
crossed the box. Drifter data is only considered one year before / after the drifter crossed the box. The bathymetry is shown in grey
with the 5000 m, 2500 m, 1000 m, 500 m, 250 m isobaths.**



## 4. Discussion

### 4.1 Circulation of Baffin Bay and Hudson Bay waters over the Labrador Shelf and into the subpolar North Atlantic

Our results show that waters found at the inner shelf near Hudson Strait—which includes Baffin Bay waters, Hudson Bay
waters, and part of the waters recirculating from the West Greenland Current—remain on the shelf until Flemish Pass as part
of the coastal and shelfbreak branches of the Labrador Current. By contrast waters originating from the West Greenland Current
and found on the outer shelf near Hudson Strait remain in the slope branch of the Labrador Current until past Hamilton Bank,
where some are steered further onto the shelf and into the coastal current. Most of the waters circulating over the inner Labrador
Shelf are exported to the subpolar North Atlantic between Flemish Pass and just past the Tail of the Grand Banks. There, they
enter the North Atlantic Current and follow its path downstream. In the following, we discuss exchanges between the shelf
and the interior regions, between the coastal and shelfbreak branches of the Labrador Current, and between the shelfbreak and
slope branches of the Labrador Current.

Consistent with previous studies (Myers et al. 2005; Penelly et al. 2019), our results show no direct connection between the
inner Labrador Shelf and the central Labrador Sea. Drifter coverage is limited in time however, and we cannot exclude the
possibility that limited and local exchanges occur, as suggested in Howatt et al. (2018) and Clément et al. (2023). One of the
drifters originating from Hudson Strait briefly entered the Labrador Sea before flowing back towards the Labrador Current,
which could be a clue of possible exchanges, albeit limited and local in nature. Further investigation of small-scale exchanges
between the Labrador Shelf and central Labrador Sea could be useful to understand local processes, however our results suggest
that an increase in freshwater input to the Labrador Shelf would primarily affect the amount of freshwater exported at the
Grand Banks of Newfoundland, rather than directly the interior Labrador Sea.

By contrast, we observe recurrent exchanges between the coastal and shelfbreak branches of the Labrador Current. Deep
troughs that cut across the shelf steer waters from the shelfbreak current towards the coast, that then mixes with the coastal
current or is steered back towards the shelfbreak.  South of 58°N, drifters that originate from Hudson and Davis Strait are not
distinguishable as they flow downstream over the Labrador Shelf. This is in contrast with the results from Florindo Lopez et
al. (2020), who argue that the coastal current stems from the Hudson Strait outflow, and that it remains distinguishable from
Baffin Bay waters until Seal Island, at 54°N. The circulation inferred from the drifter dataset is similar to early descriptions of
the Labrador Current which highlighted the influence of troughs and saddles on the circulation (Peterson et al., 1987) and
identified them as a potential source for the coastal branch of the Labrador Current (Colbourne et al., 1997). The drifter data
do not show a direct connection between the Hudson Strait outflow and the coastal branch of the Labrador Current, but more
observations would be necessary to understand this connection and the origin of the coastal branch.

Convection in the boundary current can also lead to the formation of dense waters, albeit lighter than the ones formed in the
interior Labrador Sea (Pickart et al., 1997; Palter et al., 2008). Because they can be directly exported southwards, these



intermediate dense waters were suggested to have a more direct impact on the overturning than waters formed in the interior, that first need to enter the boundary current via eddy exchange (Georgiou et al., 2019, 2021). Therefore, exchanges between

the fresh shelf and the slope branch of the Labrador Current, where the transformation of waters to denser classes is expected to happen, could impact the formation of intermediate deep waters there, and the overturning. Though the limit between the slope and shelfbreak branches of the Labrador Current is difficult to identify from our dataset, Fig 8. shows clearly different behaviours between drifters that flow over the shallower and over the deeper isobaths of the Labrador Shelf and slope, suggesting limited interactions between waters found over the shelf and at the slope. This suggests that fresh waters flowing

over the Labrador Shelf would not immediately affect the slope branch of the Labrador Current, until they reach the Newfoundland region.

**4.2 Possible impact of increasing freshwater input to the Labrador Shelf on the subpolar North Atlantic**

Our results suggest that the influence of increasing fluxes of freshwater from the Arctic, Hudson Strait and most of the west Greenland ice sheet would be limited to the inner Labrador Shelf, at least until the water reaches the Grand Banks of

Newfoundland and thus is out of the Labrador Sea. This is consistent with tracer studies focusing on the fate of freshwater released from the Beaufort Gyre (Zhang et al., 2021) and the Greenland Ice Sheet (Dukhovskoy et al., 2019; Gillard et al., 2016). As they leave the shelf between Flemish Pass and the Tail of the Grand Banks, the fresh surface waters transported by the coastal and shelfbreak branches of the Labrador Current mix with the salty and warm North Atlantic Current waters, before continuing along its path into the subpolar North Atlantic. Increased freshwater fluxes from Davis Strait, Hudson Strait and

West Greenland glaciers could therefore lead to the formation of a salinity anomaly stemming from offshore of the Grand Banks, rather than directly affect the interior Labrador Sea.

A salinity anomaly started developing offshore of the Grand Banks in 2012 (Holliday et al., 2020). The freshening was attributed to an intensified flow of Labrador Current waters off the shelf in that region following changes in regional wind patterns (Holiday et al., 2020; Fox et al., 2022; Jutras et al., 2023). Found in 2012-2016 in the Iceland Basin (Furey et al.,

2023), the salinity anomaly was then advected into the Irminger Current and the interior Irminger Sea. Bilò et al., (2022) argue that it led to a doubling of the stratification of the upper 1000 m in the Irminger Sea between 2017 and 2020 and was partly responsible for the suppression of deep convection in the winters of 2018 and 2019. The trajectory and impact of that salinity anomaly in the subpolar North Atlantic suggests that a freshening of the Labrador Shelf due to increased freshwater input from Greenland and the Arctic could also have an indirect influence on the water properties of the upper layers and on deep

convection in the Irminger and Labrador seas. The relation between increased freshwater input onto the Labrador Shelf and increased freshwater fluxes off the Grand Banks would depend on several factors and likely not be linear, making the magnitude of such an anomaly difficult to estimate. For instance, a fresher shelf would increase the density gradient between the shelf and the interior, leading to an acceleration of the Labrador Current, which could also contribute to enhancing detachment at sharp bathymetric bends (Chapman, 2003; Jutras et al., 2023).





How increased freshwater input to the subpolar North Atlantic may affect the region is dependent on the origin of that freshwater: Freshwater that first circulates over the East Greenland shelf, originating from Fram Strait or from Greenland, can directly enter the Labrador Sea via the West Greenland shelf, and therefore potentially have a direct impact on the salinity and stratification of its upper layer. This is exemplified by the 1970s great salinity anomaly which was attributed to increased Fram Strait outflow (Dickson et al., 1988; Hakkinen et al., 1993) and was suggested, together with reduced surface heat fluxes, to

have contributed to a temporary shutdown of deep convection in the Labrador Sea (Lazier et al., 1980; Gelderloos et al., 2012). On the other hand, freshwater that stems from Baffin Bay and Davis Strait, or from most of the West Greenland ice sheet, can only enter the interior subpolar North Atlantic further downstream: at the Grand Banks of Newfoundland. It then also has the potential to affect deep convection regions, but indirectly, as a vertically mixed salinity anomaly that circulates around the subpolar North Atlantic on a timescale of several years. Moreover, different processes that can lead to increased freshwater

input to the subpolar North Atlantic from different regions operate at different timescales: While a release of the Beaufort Gyre could lead to several thousands of cubic kilometres of waters entering the region within a few years (Zhang et al., 2021; Timmermans and Toole, 2023), the accelerating melt of the Greenland Ice Sheet is a more continuous but also slower process.

## 5. Conclusion

This study presents an investigation of the circulation of fresh waters originating from Baffin Bay and Hudson Bay over the

Labrador Shelf and into the subpolar North Atlantic, based on a surface drifter dataset spanning 1990 to 2023. It is motivated by the need to understand how increasing freshwater input from Greenland and the Arctic may affect the subpolar North Atlantic in the coming decades, and its impact on key processes such as deep convection in the Labrador Sea.

We show that waters originating from Baffin / Hudson Bay mostly remain on the Labrador Shelf before they reach the Newfoundland Shelf, and do not directly enter the convection region of the Labrador Sea. They are only exported to the

subpolar North Atlantic downstream of Flemish Pass, that is past the Labrador Sea. There are also only limited exchanges between the shelf and slope region of the Labrador Current, suggesting no direct influence of these fresh waters on convection in the boundary current.

Using the drifters, we define typical pathways between Baffin / Hudson Bay and the interior subpolar North Atlantic: Baffin Bay waters first flow over the Baffin Island shelf via the Baffin Island Current. They then follow the topography to the mouth

of Hudson Strait, and are joined by waters from Hudson Strait and the recirculating West Greenland Current. On the Labrador Shelf, most of these waters are steered towards the coast by deep canyons. Part remains in the coastal current downstream, while the rest re-enters the shelfbreak Labrador Current. The major part of Baffin / Hudson Bay origin waters found over the Labrador Shelf then flows through Flemish Pass, and most are retroflected between the southern end of Flemish Pass and the tail of the Grand Banks. The waters exported at the Grand Banks mix into the North Atlantic Current, and spread in the subpolar

North Atlantic along its path.





Increasing freshwater input to the Labrador Shelf from Baffin Bay and Hudson Bay would likely not directly affect the central Labrador Sea, but rather could possibly lead to a freshwater anomaly stemming at the Grand Banks and circulating around the subpolar North Atlantic. This freshwater could then indirectly affect deep convection regions and in turn ventilation in the subpolar North Atlantic, as a weakened and vertically mixed anomaly. These results highlight the importance of considering

the different possible origins of freshwater in the subpolar North Atlantic, as fresh waters originating from Fram Strait, Davis Strait and Greenland follow different pathways, and are influenced by different parameters, impacting where, how and how fast freshwater may enter the interior.






**Code and data availability**

All data used in this study is publicly available.

The EGC-DrIFT dataset is available on the NIOZ dataverse repository (https://doi.org/10.25850/nioz/7b.b.ff, Duyck and De

Jong, 2023b).

The GDP dataset is available from NOAA (https://www.aoml.noaa.gov/phod/gdp/interpolated/data/all.php , Lumpkin and Centurioni, 2019). We used the ASCII files, updated through May 2023 (last accessed [2023-10-12]).

The bathymetry is the ETOPO2022 60 arc-second ice surface elevation, available from NOAA at https://doi.org/10.25921/fd45-gt74 (NOAA, 2022).

The scripts written to analyse the data and produce the figure are available at https://doi.org/10.5281/zenodo.13255612 (Duyck, 2024)

To perform the analysis and produce the figures, we made use of the toolboxes M_Map (Pawlowicz, 2020*) and jLab (Lilly 2024).

**Supplement**

The supplementary figures are available online at: (waiting for doi)

**Author contributions**

ED and EFW conceptualized the analysis, ED designed the methodology and performed the analysis. ED wrote the manuscript, with advice and critical feedback from EFW and NF. All authors discussed the results and finalized the paper.

**Competing interests**

The authors declare they have no conflict of interest

**Disclaimer**

Publisher's note: Copernicus Publications remains neutral with regard to jurisdictional claims in published maps and institutional affiliations.

**Acknowledgements**

We thank the captains, crews and science teams of all the research cruises that deployed the surface drifters we used in this analysis.





**Financial support**

ED and EFW received funding from the European Research Council (ERC) under the European Union's Horizon 2020 research and innovation programme (grant agreement No 803140). NF was supported by grant 2123128 from the National Science
Foundation. Views and opinions expressed are however those of the author(s) only and do not necessarily reflect those of the European Union or National Science Foundation. Neither the European Union nor the National Science Foundation can be held responsible for them

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
