# Peer review of "Circulation of Baffin Bay and Hudson Bay waters on the Labrador Shelf and into the subpolar North Atlantic"

_EGUsphere, 2024_

## Author Response (AR1)

Reviewer 1

*This article examines the export of ocean drifters from the continental shelf in the Northwest Atlantic to the Labrador Sea and subpolar gyre. The goal of the article is to examine potential freshwater pathways from a variety of Arctic sources such as Hudson Bay, Davis Strait, Baffin Bay, and Greenland ice sheets. The authors use Surface Velocity Program (SVP) drifters which have a long holey-sock drogue centered at 15m depth. The authors find that, of the drifters analyzed, most are exported off the shelf at a few key locations. For drifters originating from Davis Strait and Hudson Bay, the main export region was near the Grand Banks on the Newfoundland Shelf. They conclude that fresh water originating from Davis Strait and Hudson Bay is unlikely to have a direct impact on the Labrador Sea and deep convection because it would be entrained into the North Atlantic Current and subpolar gyre before entering the Labrador Sea.*

*The article is well-written and covers scientific questions that are of interest to the readership of Ocean Sciences, that is, what is the fate of fresh water from the Arctic and what is its connection to the Labrador Sea. The application of observed drifter data is a novel approach in addressing this question, however, I would argue that there are limitations when using this dataset. Those limitations will be described in more detail below. For that reason, I believe some of the conclusions are too strong given the analysis performed and I would recommend a major revision.*

*Below, I have provided some specific comments that, once addressed in a revised manuscript, would lead to an improvement. I would also be open to considering a rebuttal if the authors disagree with my position.*

We thank the reviewer for their comments and suggestions.

We agree that there are limitations to the use of the drifter dataset, both due to the nature of drifter data, and to the paucity of observations in the region studied. We tried to make these limitations clearer throughout the paper. In the methods section, further indications about the limitations of the drifter dataset were added. These caveats and how they may influence our results were also explicated further in the conclusions. We also added a paragraph to the discussion, as to expand considerations on potential exchanges we would not be able to observe using the drifter dataset, and on how our results complete existing studies by providing an additional observational perspective to a topic mostly explored using modelling studies. We expand more on this in the responses to specific comments below. Several changes were also made to the text and figures following suggestions from reviewer 2, details can be found in that part of the responses.

*1. Sparse data and sampling biases – Drifter tracks are very chaotic. For instance, two drifters of the same type deployed at the same place and time will eventually separate, especially when considering the spatial and temporal time scales discussed in this study. So, a small group of drifter tracks cannot represent all the possible pathways between areas of interest. A large sample size is needed. There are only 11 drifters that originate from Davis Strait, so statements like "On the other hand, freshwater that stems from Baffin Bay and Davis Strait, or from most of the West Greenland ice sheet, can only enter the interior subpolar North Atlantic further downstream: at the Grand Banks of Newfoundland" (Lines 461-462) are not supported.*

*I appreciate the authors' ingenuity and efforts to increase the sample size by including drifters introduced to the system downstream of the main fresh water sources considered. This is a nice dataset, and it is encouraging to see scientists apply this data to investigate important questions. My main concern, however, is that the conclusions are presented without an adequate description of the limitations of the data and methods.*

Thanks for these comments. We agree that it is not possible to identify all possible freshwater pathways using the drifter dataset. We clarified that the goal of the study is rather to characterize main pathways from Baffin and Hudson Bay to the interior Subpolar North Atlantic (*lines 78-79*), using a newly available dataset in a region where observations are limited (in particular for the northern part of the Labrador Shelf, *lines 120-121*): "*The goal of this study is to characterise the pathways followed by freshwater originating from Baffin and Hudson Bay on the Labrador Shelf and into the subpolar North Atlantic*" ; "*Despite these limitations, the drifter dataset can provide useful additional results in a poorly sampled area, and complete existing observational and modelling studies*"

We expanded existing discussions of limitations of the drifter dataset. In section 4.1, the paragraph discussing exchanges between the shelf and the interior was divided in two:

The first part now discusses how the results obtained using the drifter dataset offer a useful observational counterpart to existing modelling studies that investigated the fate of surface Labrador shelf waters and also found limited to no exchanges between the shelf and interior Labrador Sea. Several examples of such studies are now mentioned in that paragraph (Myers 2005, Gillard et al 2016, Dukhovskoy et al 2021, Pennelly et al 2019, Schulze-Chretien and Frajka-Williams 2018, *lines 427-434*), and we emphasize that our results complete these studies : "*While the drifter dataset is concentrated in time, highly biased towards the summer season, and cannot capture all possible pathways for fresh surface waters originating from Davis Strait and Hudson Strait, it offers a useful observational counterpart to modelling studies, and mostly comforts their results.*", lines 434-436.

The second part focuses on the role of small-scale exchanges and discusses the role of winds, considering types of exchanges that might not be captured by our dataset. We expand further on the changes made to this paragraph in the response to the following comment.

We also clarified in several parts of the text that the results discussed are specifically derived from the dataset we are using in the study, and put more emphasis on uncertainties (For instance, in the conclusion, lines 514-517: "*The drifter dataset comes with several caveats: it is concentrated in time, limiting investigation of interannual variability, and it is biased towards the summer and autumn seasons. Despite these limitations, the results presented here help bridge an important gap in observational studies focused on freshwater pathways in the Labrador shelf region.*"; and lines 532-533: "*Our results suggest that an increase in freshwater input to the Labrador Shelf would primarily affect the amount of freshwater exported at the Grand Banks of Newfoundland, rather than directly the interior Labrador Sea*").

The sentence mentioned by the reviewer was rewritten, using "mostly" rather than "can only" (now line 500), and similar changes were made throughout the text to emphasize uncertainties.

*2. Sampling depth of drifters and representativeness of fresh water – After reading this article, I am left wondering how representative these drifters are of freshwater pathways. The drifters are drogued at 15m and so they represent near-surface currents, however, I am wondering how deep the fresh water extends in these areas and if there is any vertical shear in the currents. For instance, would fresh water closer to the surface take a different path or, similarly, would drifters drogued at a shallower depth give the same result? I recognize the authors are working with the data that is available and a comparison of between the SVPs and other drifter types is not possible. Perhaps though, a description of the seasonal stratification could help the readers interpret the results.*

*Furthermore, fresh water mixes laterally. Drifters at the shelf edge or meandering back and forth across the export boundary may indeed represent a flux of fresh water from the shelf to the interior oceans through mixing. Even though these fluxes might be much smaller than other sources, like*

*eddies shedding from the West Greenland Current, they might still be important as suggested in Clements et al., 2023. The drifter data analysis presented in this article does not account for this type of exchange.*

We thank the reviewer for these remarks.

Regarding the sampling depth of drifters and possible exchanges that would not be observable using this dataset: We now explain in the methods that drifters do not directly allow to define freshwater pathways, but instead help characterize the surface circulation, here at 15 m depth (*Lines 119-120: "While the SVP drifters allow inferences of the ocean circulation at 15 m depth and can give an indication of freshwater pathways, they do not directly track freshwater"*). We also added a short discussion of the importance of the depth at which drifters are drogued in the discussion (Lines 446-449: *"The potential to observe such exchanges might also be affected by the drogue depth of the drifters, which are all anchored at 15 m in our dataset: Using drifters anchored at 15 m and 40 cm depth, Duyck and De Jong (2021) showed that the shallower drifters were more easily driven off-shelf at Cape Farewell during export favourable wind events."*).
Unfortunately, none of the 40 cm depth drifters used in that study were found on the Labrador Shelf, so we cannot replicate this analysis in that region. We also added some information in the introduction about the seasonal variations of salinity at the surface Labrador Shelf, which would indeed affect how much freshwater can be driven off-shelf by cross shelf exchanges (lines 65-68*: The salinity of waters transported in the Labrador Current is driven by seasonal variations in river discharge and ice melt, leading to a strong seasonality, with peak freshness of surface waters observed at the Labrador shelf in early summer, and reaching the newfoundland shelf in late summer (Fratantoni and Mc Cartney, 2010)".*

Regarding lateral exchanges: We expanded on this topic in the discussion, section 4.1. both acknowledging that the dataset and analysis used in the study, do not allow to specifically investigate time limited, local exchanges: "*The present results do not however exclude the occurrence of limited exchanges between the shelf and the interior, such as suggested in Howatt et al. (2018) and Clément et al. (2023), which could be missed by the drifter dataset* ", lines 439-440. We also suggest possible further studies to understand for instance the role of winds in driving these. *"Further studies investigating such exchanges could in particular focus on the role of winds in constraining or enhancing them",* lines 443-444. *"While winds over the Labrador Shelf are predominantly downwelling favourable (Yang and Chen 2021), and contribute to constraining surface waters towards the coast, Howatt et al (2018) identified freshwater export off the Labrador Shelf south of Hamilton Bank during upwelling favourable winds",* lines 444-446.

*3. Confusing and contradictory statements – The article contains some statements that are unclear and confusing. For example, Lines 309 – 310 – "By contrast, all the drifters that at some point circulated over the shelf were driven into Flemish pass (in blue), with a small minority going over Newfoundland shelf (in green)." How can all drifters go into Flemish Pass while at the same time a minority go over the Newfoundland shelf?*

*Furthermore, lines 254-256 state that Hudson Strait/Davis Strait waters are only exported downstream of Flemish Pass contradicts the previous paragraph which describes at least one of the Hudson Strait/Davis Strait drifters crossing the Labrador Shelf boundary. While I understand that drifter remains in the Labrador Current, it shows there is potential for freshwater transport from Hudson Strait into the Labrador Sea through a combination of advection and lateral mixing, as*

*discussed in a previous comment. Given the small sample size of Hudson Strait and Davis Strait drifters, these types of pathways may simply be under sampled with this dataset.*

*I encourage the authors to clarify these statements and avoid using absolutes like "all" or "only", unless the analysis has supported their use.*

We thank the reviewer for their comments. We agree that these statements were contradictory, and tried to make them more consistent.

We rewrote the sentence that was lines 309-310 (now lines 330-332), which main point was to highlight that none of the drifters that originated in the coastal current flowed around Flemish Cap. It now says: *"Fig. 8f suggests that the drifters that circulated around Flemish Cap remained offshore of the Labrador Shelf before reaching that region (trajectories in green), while the drifters that at some point circulated over the shelf were either driven into Flemish pass (in blue), or flowed over the inshore Newfoundland shelf (in red)."*

As mentioned by the reviewer, one of the drifters indeed crosses the boundary, re-entering it further downstream. We rewrote this sentence to clarify that no drifter (rather than "waters") was exported before Flemish Pass. It now says: *"the trajectories of drifters originating from Davis / Hudson Strait suggest little direct connection between Baffin / Hudson Bay and the interior Labrador Sea. These drifters only leave the shelf downstream of Flemish Pass", lines 272-274.* This does not exclude that a drifter coming in and out of the shelf could be a signal of surface water exchange in that region. The behaviour of this drifter and its meaning for surface water exchanges is also discussed in section 4.1, lines 441-448.

Throughout the paper, we softened the tone of the sentences, including words such as "mostly" or "primarily" to emphasize uncertainties in the analysis. We also specified where necessary that the behaviour discussed is specifically from the drifters included in our dataset.

*4. Inconsistent naming conventions – I find the naming conventions, especially for the "Newfoundland boxes" first mentioned in Figure 3, to be a little bit confusing and hard to follow. Specifically, in the text that describes Figure 7 on lines 297-310, the boxes are referred to as "Newfoundland box", "Flemish Pass box" and "Flemish Cap box", but in Figure 7 they are "Newfoundland 1", Newfoundland 2", and "Newfoundland 3". It would be clearer to use the same naming in the text and figures. A similar comment applies to the Labrador Shelf boxes. In the text they are referred to as inshore and offshore but in Figure 7 numbers are used to label the boxes.*

We thank the reviewer for pointing out these inconsistencies in our naming conventions. We revisited the names we use and adjusted the figures accordingly: We now use "origin", "northern Labrador Shelf", "mid- Labrador shelf", "southern Labrador shelf" and "Newfoundland" as section names, and name boxes according to their being "onshore" of "offshore", or to the topographic features they are drawn at ("Davis Strait", "Flemish Cap", etc)

*5. Fate of the exported drifters – In several places, the article describes that the drifters exported from the shelf eventually end up in the subpolar North Atlantic (e.g., Lines 484-485). These statements are misleading because Figure 10 shows evidence of some of the drifters entering the subtropical North Atlantic.*

Though this is not very visible in Figure 10, which shows the main regions where drifters flow after passing the Flemish Cap / Flemish Pass boxes, a few drifters do indeed exit the North Atlantic Current

to the south. This happens for both the Flemish Cap and Flemish Pass drifters. One of the Flemish Cap drifters enters the subtropical gyre. The fate of the other drifters that flow south of the North Atlantic Current is unclear (their trajectory stops in the middle of the Atlantic, likely as they stop transmitting data or lose their drogue).

The majority of the drifters originating from Flemish Cap and Flemish Pass flow north-eastwards into the Subpolar North Atlantic with the North Atlantic Current. Here too, and for the same reasons, their ultimate fate is not clear, though a few drifters are visibly recirculating into the Irminger Current and the East Reykjanes Ridge Current. These results are similar to the trajectories shown in the Figure 5 of Reverdin et al (2003), which shows drifters deployed north of the North Atlantic Current mostly remaining in the Subpolar North Atlantic, with a small minority heading towards the Subtropical North Atlantic.

We adapted sections that mention the ultimate fate of drifters to better reflect the full trajectories of these drifters. In particular, we modified the sentence referred to in the comment (now line 531-532: *"The waters exported at the Grand Banks mix into the North Atlantic Current, and spread towards the subpolar North Atlantic along its path."*), and expanded on the further fate of drifters in the following paragraph, which also refers to the propagation of past salinity anomalies from that region. Similarly, lines 401-403, we now write: *"Most of these drifters then remain in the Subpolar North Atlantic, with a few leaving the North Atlantic Current to the south, possibly heading into the Subtropical Gyre*. "

*6. Line 94 – Could you provide a brief description regarding how the velocities are computed?*

We now explain on line 102 that the velocities are computed from the filtered drifter trajectories using a central difference algorithm, which gives the average of a forward and backward difference algorithm.

*7. Line 110 – I am curious about how you determine a drifter is caught in sea-ice. Can you provide a brief description?*

We used the EUMETSAT OSISAF sea ice concentration dataset to verify whether drifters might have been affected by sea ice as they flowed down the Labrador shelf. Focusing on drifters deployed at Davis Strait in September 2021, we observed that starting from December 2021, the sea ice expanding on the Labrador shelf started catching up with drifters flowing over the shelf. As the sea ice caught up with them, these drifters either stopped transmitting, or lost their drogues. We included a figure in supplementary materials to show this process via three snapshots (Figure S2), referred to it line 118, and mentioned the sea ice concentration dataset in the data availability section.

*8. Line 187 – I am not sure what is meant by "core". Could you be more descriptive?*

We clarified that we were referring to the one of the cores of the Labrador Current by using "current core".

**Minor comments and typos**

1. *Figure 1b - The duplicate copies of 30' on the y-axis in the figure is confusing because the degrees for those gridlines are not provided and substituting the degrees from a nearby gridline doesn't make a lot of sense. From what I can gather, these two gridlines are 52 degrees 30' and 57*

*degrees 30'. Is that right? Is it possible to provide the degrees for those two gridlines so that readers do not have to guess?*

The labels were adjusted to indicate 52.5°N and 57.5°N

2. *Figure 4 – the units on the colorbar are m/s. I believe this is a mistake and the units should be cm/s because 60 m/s currents are not observed in this area (if anywhere!).*

Indeed, thank you for spotting the mistake. This was adjusted and the legend now says "cm/s"

3. *Figure 5a – The caption describes that a fraction is plotted, but the colorbar describes the number of drifters. This discrepancy should be corrected.*

The caption was corrected to indicate that the colors indeed correspond to the total number of drifters exported at a given segment.

4. *Figure 8a – The orange track is hard to see. The plot could be clearer by using a thicker line.*

This figure was remade, and we are now using thicker lines for these tracks.

5. *Figure 10 – The axis titles are not consistent with what is described in the caption. The titles indicate Flemish Cap is on the left and Flemish Pass is on the right, but the caption says the opposite. Please correct this discrepancy.*

Thank you – we have now corrected this discrepancy.

6. *Figure 10 – Why are there blue data areas south of Flemish Cap in the plot on the right?*

This was an artefact from not limiting the selected drifters to those that originate from the Labrador Sea and Labrador Shelf. A few drifters that flowed with the North Atlantic Current recirculated in the region offshore of Flemish Cap passed through the Flemish Cap box and were shown in the plot. Now, only drifters that passed through both the southern Labrador shelf section and the Flemish Cap box are shown in the plot.

7. *Figure S1b – This figure is difficult to interpret for several reasons. First, because some of the grey tracks are easily confused with the bathymetry contours. Second, the dotted blue track does not appear to have a solid counterpart which makes it difficult to determine when the data gap occurred. Finally, the caption mentions the red track is from Clements et al., 2023. Is it possible to specify the WMOID of the relevant drifter track? When I looked at the drifter tracks mentioned in the paper, neither of them have data as far north was what is represented by the orange-red solid line in this plot.*

The grey trajectories corresponded to drifters that present such gaps but did not circulate over the Labrador shelf. These are not essential to understand how we filtered the data, and we have now removed them from the figure for clarity. The blue trajectory corresponds to a drifter for which the data gap occurred before it reached the region. This is now explained in the legend.

The reference to Clement et al. (2023) was to their figure 7, but indeed, there was a mistake, the drifter we did not include in the study is not the one shown in that figure. The drifters from Clement et al. (2023) are included in this study. They are offshore of the 1000 m isobath we denote as the shelfbreak and flow with the slope current, which shows exchanges with the interior.

8. *Figure S3 – this figure needs an improved caption, and some revision. What do the orange arrows represent? Why is Davis Strait still labelled at the top if the figure is means to represent Hudson Strait drifters?*

Additional information was added to the caption. The figure is very similar to Figure 7 and does not represent Hudson Strait drifters but rather also shows results from the box to box pathways analysis, with less boxes. This was done to confirm that the results are consistent when using a different number of boxes. The confusing caption was due to a change in the box naming during the writing of the paper, which has now been fixed.

9. *Line 62, 208, 418 - Colbourne et al., 1997 is not listed in the References.*
10. *Line 65 – Petri and Anderson 1983 should be Petrie and Anderson, 1983.*
11. *Line 89, 91 – The citations to Duyck and De Jong, 2023 need to be clarified because the reference list contains an a and b.*
12. *Line 127 – "corresponding to 750km" – if the segments are 200km apart, wouldn't 5 segments apart correspond to 1000km?*
13. *Line 133 – "divide the region into five section." – section should be sections*

We thank the reviewer for pointing out these errors in the original submission, and we have now corrected them.

14. *Line 171 – "along the 1500m" – the caption in Figure 4 indicates that the 1000m isobath is plotted. Given this paragraph is discussing Figure 4, it would be clearer if the same isobath is referenced in both the text and figure.*

We modified the text and now mention that this branch of the West Greenland Current flows between the 1000 m and 2500 m isobaths, which is more clearly identifiable on the figure. The steep slope in that region makes the 1000 m and 1500 m isobaths very close to each other.

15. *Line 107 – "Funk Island Deep" is not labelled in Figure 1, rather Funk Island Bank is. Is it possible to label Funk Island Deep for the readers who are unfamiliar with the region?*

This was added to the figure.
* * *
*This paper looks at the circulation of low salinities water on the Labrador Shelf and explores the origins of said waters, focussing on those waters from Hudson Bay and Baffin Bay. The authors use multiple drifter data sets covering 1990-2023 to examine these questions. Although there are some data sparsity issues, the authors are able to pull together a very complete analysis. The authors shows that waters from the northern straits remain on the Labrador shelf, with offshore exchange beginning on the Newfoundland Shelf. Different pathways from these waters are shown, including that there is significant exchange between the inshore and shelfbreak component of the Labrador Current. Thus, the authors conclude, that freshwater anomalies in the Labrador Current will initially be transported to the area off the Grand Banks, rather than directly to the convective regions of the Labrador Sea.*

*This is a timely and well written paper, topical and adding novel data to the question of freshwater pathways in the western Labrador Sea. The figure quality is generally high as well. Thus, this paper will be an excellent addition to the literature. I would suggest minor revisions, with details provided below.*

We thank the reviewer for their comments, that allowed to improve the manuscript.

Based on these and comments from Reviewer 1, we added more discussions of the limitation of the dataset throughout the manuscript and specifically in the discussion section.

Following the reviewer's suggestion, we added an additional box for the section to section analysis, to capture drifters originating from the West Greenland current. We also slightly re-drew the boxes, both to make them more visible and so that they are defined more consistently (in particular all the offshore boxes now extend to 100m off the 1000m isobath). This led to more drifters being included in the analysis (but most off-shelf), and to slight changes in absolute numbers of drifters flowing from section to section, but the results remain very similar.

Following suggestions, several changes were also made to figures, which improved their readability. In particular, in Figure 8 the trajectories of drifters flowing onshore are now drawn last, to make them more visible. Figure 7 was redesigned, and the names of the boxes are now consistent with the text. Figures 2 and 4 now use a discrete colormap, which makes their interpretation easier.

Further details can be found in the response to specific comments below.

*Data density: The authors use the available drifter data and do a good job putting it together. Yet, figure 2 does show most of the data is from off the shelf, while the analysis focusses on freshwater on the shelf. I think some greater discussion of this limitation, especially in the discussion, would be good. As an aside, part of me wishes this analysis could be augmented with model virtual float data, in a joint study/paper, rather than seeing the observational and modelling work in separate papers (I know this is beyond the authors scope for this paper, but I still want to make the comment for long-term thought).*

Most of the available drifter data is indeed off the shelf, and the data available on the shelf is rather concentrated in time, with most data between 2020 and 2023. Further discussion of the limitations of the drifter dataset were added to the methods and results sections. In addition, we added a paragraph in the discussion (section 4.1) to both discuss the limitations of the dataset we are using, and connect our results to existing modelling studies. A few such studies indeed already investigated the fate of Greenland and arctic freshwaters using notably virtual particles and passive tracers, and we discuss how our results, even if based on a sparse dataset, allow to provide a useful observational counterpart to these studies.

*Bathymetry: At multiple places, the authors refer to the bathymetry. This incudes in figures, such as number 1 and 2. And in there calculations when they define various boxes related to water depth. Yet I don't see any reference, anywhere in the manuscript, for the source of this topographic data. It needs to ben mentioned and reference. As well, given the potentially issues with bathymetric quality in this region, might the authors want to add some discussion of whether there results are sensitivity to uncertainties in the data.*

The information about the bathymetry is provided in the code and data availability section (lines 549-550), and there is a reference to it in the reference list (NOAA 2022).

The main feature we use for our analysis is the smoothed 1000m isobath, that symbolizes the shelfbreak. This is a very steep topographic feature, which is also visible in satellite altimetry derived bathymetry. Other features, such as the canyons that cut through the Labrador shelf could be more subject to uncertainties. While these distinctions are worth noting, we do not believe they affect the main findings of the paper.

*The section/box definitions: The authors divide their study region north-south using 5 sections. Those seem logical choices. They also discuss using 10 boxes, to divide up the sections into east-west definitions as well. This is where I am a bit confused. Have they divided the area between sections into two long boxes, covering the inner and outer shelf? Yet the pink boxes in figure 3 are small – are they then the corners of the boxes? Yet, some sections have different number of pink boxes. So, I will admit to some confusion. This needs to be more clearly explained. And ideally, the actual 2D boxes need to shown on some figure. Additionally, given the authors do at least discuss waters inflowing from the West Greenland Current, why no sections/boxes on that side of the Labrador Sea? It might also be interesting to have a north-south section across the mouth of Hudson Strait, with some division for the inflowing and outflowing components.*

We thank the reviewer for these comments. Based on these, we made several adjustments to the definition of the boxes, described below.

West Greenland Current box: We initially did not include such a box given the focus on drifters circulating over the Labrador Shelf. After testing the addition of such a box, we however found that it is indeed quite helpful to clarify the origin of the drifters flowing off the shelfbreak. We thank the reviewer for this suggestion.

Box definition: The boxes were redrawn both to make them more visible on the figures, and to make their definition more consistent. In particular, all offshore boxes are now extended until 100km off the 1000m isobath. This re-definition of boxes leads to changes in the absolute numbers of drifters considered in the analysis, and flowing from box to box, but the results are very similar to the ones obtained with the previous box definition. Figure 8, we also changed the order in which trajectories are plotted on the figure to improve their readability, by plotting the most onshore trajectories, which have the least number of drifters, last.

Figure 3b: We remade this figure based on the reviewer's comments. The pink lines indeed defined the corners of the boxes. The full boxes are now shown on the figure, coloured depending on the section they are part of.

Hudson Strait box: Before choosing the sections used in the study, we tested a similar method with about twice more boxes. We removed the boxes that did not allow to infer additional information. The Hudson Strait box was among these, as all drifters flowing through that box also flowed through the inshore box of the Northern Labrador Shelf section (But not all of these flowed through the Hudson Strait box). As a side note, a large part of the drifters exiting Hudson Strait actually originated from Davis Strait and flowed in and out of the Hudson Strait. It could therefore be quite interesting to use these drifters to study the in and out flow across Hudson Strait, but this is beyond the scope of this paper.

*Quantitative Freshwater Export Estimates: Given the focus on freshwater, can you make any estimate of freshwater transport by the floats, and/or offshore in the different regions? This would be of value in comparing this work with the results from other analyses and approaches.*

The drifter data does not allow for such a quantification, and it is therefore beyond the scope of the current study, based on the drifter dataset. The results presented in this study could however be very useful as a reference for further analyses trying to make such quantifications based on model or remote sensing results.

*Line 30: Is there any evidence for weaken of deep water formation yet? The paper would benefit from a short discussion of this topic.*

We included a sentence lines 36-38 "there is no agreement on the volume of freshwater that would need to enter these regions to affect deep convection, whether Greenland freshwater is already affecting salinity and deep convection, nor how far in the future this could occur (Böning et al., 2016; Yang et al., 2016; Dukhovskoy et al., 2019)" to refer to the absence of agreement on whether Greenland freshwater might already be affecting deep convection in the Labrador Sea. We do not consider the time variability of convection in this paper because it is beyond the scope of the research. However, we agree with the reviewer that it is certainly a topic that deserves more scrutiny, especially in light of the findings we report here.

*Lines 43, 46: I see at least 2 mispelled author surnames, in different papers – Saucier and Pennelly.*

Thank you, we have now corrected these names.

*Line 46: For the Hudson Bay origin, might Florindo-Lopez et al also be a good reference to add here.*

Florindo-Lopez et al (2020) is referred to further in this section. We refer to the paper from Straneo and Saucier here because it investigates the inflow and outflow of Hudson Strait specifically, while the Florindo Lopez et al paper focuses more on the downstream pathways of Hudson Strait waters downstream. We therefore cite it when mentioning that topic (lines 61-63: *"While the coastal Labrador Current can be distinguished as a separate branch, its origins are disputed, with literature both suggesting it stems from the Hudson Strait outflow (Florindo Lopez et al., 2020)"*)

*Figure 2 shows 2 coloured boxes, but I don't see them defined in the caption. I would also like to see the time period mentioned in the caption to remind the reader. Also I have trouble seeing the grey bathymetry lines. A different colour would help.*

The caption now explains that the colored boxes correspond to the Labrador shelf and Labrador Sea regions used in the histograms in the other panels of the figure. We also added the time period covered by the drifter dataset used in the study to the caption.

We agree the grey bathymetry lines were not easy to distinguish on the original figure. The color scale of the figure was changed, from continuous to discrete, which might also help with distinguishing the bathymetry. Though the grey lines are not ideal, other colours unfortunately didn't work well either because of the strong color contrasts between the interior Labrador Sea and the shelf on that figure, (due to strong contrasts in the amount of available data in each region.

*Figures such as 2 and 4 could use discrete colour bars, like in figure 3 (and others).*

Thank you for this suggestion. We changed both figures to discrete colour bars, which improved the readability of the figures.

*Line 95: Saglek Bank*

Thanks, we corrected the mistake.

*Line 215: The Flemish Pass section is close to the 47N section studied in detail by the group from the University of Bremen. Might be useful to bring in and reference some of that literature.*

Thank you for this suggestion. The Flemish Pass section is indeed close to the 47N section. However much of the focus of these studies is on the deep western boundary current, its interaction with the North Atlantic Current, and the propagation of Labrador Sea Water, while we are focusing on the very surface layer.

*Figure 6: It looks like some of the Labrador Curent trajectories loop back on themselves. Is that related to the floats entering/leaving eddies? Do you see eddies propagating down the shelf break in the data set?*

Thanks for the suggestion. This is indeed interesting, but we have not specifically investigated this. Further studies could use the drifters in combination with satellite data to investigate eddies propagating down the shelfbreak, and whether these may also play a role in exchanges between the shelf and interior Labrador Sea.

*Discussion: I think the paper could be strengthened with some suggestions of future studies, based on gaps or limitations in the present work. At the very least, it sounds like application of additional drifters in the northern Labrador Sea would likely be of value.*

Caveats and limitations to the present study were added to the methods, discussion and conclusion sections. For instance, in the conclusion, lines 522-524*: The drifter dataset presents several caveats: It is concentrated in time, which prevents investigating the role of interannual variability, and is biased towards the summer and autumn season.*

We also added a few suggestions for future studies to the discussion and conclusion. Though further deployments of drifters could be useful, some important caveats would remain even with a higher data density, such as the difficulties to gather data outside of summer months and the time concentration of the dataset. We therefore focused on the suggestion to investigate the role of winds and sea ice in preventing or enhancing exchanges, which is a blind spot of the current study. We also suggested aiming for a better understanding of the different possible influence of freshwater from different origin in the subpolar north Atlantic in future studies. For instance: *"Further studies investigating such exchanges could in particular focus on the role of winds in constraining or enhancing them",* lines 443-444; *"Further studies could focus on understanding how all these possible freshwater pathways may differently affect the future of deep convection in the Subpolar North Atlantic.",* lines 509-510